# LEARNING TO ORGANIZE KNOWLEDGE WITH N-GRAM MACHINES

## ABSTRACT

Deep neural networks (DNNs) had great success on NLP tasks such as language modeling, machine translation and certain question answering (QA) tasks. However, the success is limited at more knowledge intensive tasks such as QA from a big corpus. Existing end-to-end deep QA models (Miller et al., 2016; Weston et al., 2014) need to read the entire text after observing the question, and therefore their complexity in responding a question is linear in the text size. This is prohibitive for practical tasks such as QA from Wikipedia, a novel, or the Web. We propose to solve this scalability issue by using symbolic meaning representations, which can be indexed and retrieved efficiently with complexity that is independent of the text size. More specifically, we use sequence-to-sequence models to encode knowledge symbolically and generate programs to answer questions from the encoded knowledge. We apply our approach, called the N-Gram Machine (NGM), to the bAbI tasks (Weston et al., 2015) and a special version of them ("life-long bAbI") which has stories of up to 10 million sentences. Our experiments show that NGM can successfully solve both of these tasks accurately and efficiently. Unlike fully differentiable memory models, NGM's time complexity and answering quality are not affected by the story length. The whole system of NGM is trained end-to-end with REINFORCE (Williams, 1992). To avoid high variance in gradient estimation, which is typical in discrete latent variable models, we use beam search instead of sampling. To tackle the exponentially large search space, we use a stabilized auto-encoding objective and a structure tweak procedure to iteratively reduce and refine the search space.

## 1  INTRODUCTION

Although there is a great deal of recent research on extracting structured knowledge from text (Dong et al., 2014; Mitchell et al., 2015) and answering questions from structured knowledge stores (Dong & Lapata, 2016; Jia & Liang, 2016; Liang et al., 2017), much less progress has been made on either the problem of unifying these approaches in an end-to-end model or the problem of removing the bottleneck of relying on human experts to design the schema and annotate examples for information extraction. In particular, traditional natural language processing and information extraction approaches are too labor-intensive and brittle for answering open domain questions from large corpus, and existing end-to-end deep QA models (e.g., (Miller et al., 2016; Weston et al., 2014)) lack scalability and the ability to integrate domain knowledge.

This paper presents a new QA system that treats both the schema and the content of a structured storage as discrete hidden variables, and infers these structures automatically from weak supervisions (such as QA pair examples). The structured storage we consider is simply a set of "n-grams", which we show can represent a wide range of semantics, and can be indexed for efficient computations at scale. We present an end-to-end trainable system which combines an text auto-encoding component for encoding knowledge, and a memory enhanced sequence to sequence component for answering questions from the encoded knowledge. We show that the method scales well on artificially generated stories of up to 10 million lines long (Figure 4). The system we present here illustrates how end-to-end learning and scalability can be made possible through a symbolic knowledge storage.

## 1.1 QUESTION ANSWERING AS A TESTBED FOR TEXT UNDERSTANDING

We first define question answering as producing the answer $a$ given sentences $\mathbf{s} = (s_1, \ldots, s_{|\mathbf{s}|})$ and question $q$. Each sentence $s_i$ is represented as a sequence of words, i.e. $s_i = (w_1^i, \ldots, w_n^i)$. And the question $q$ is also represented as a sequence of words, i.e. $q = (w_1^q, \ldots, w_m^q)$. We focus on extractive question answering, where the answer $a$ is always a word in one of the sentences.

Despite its simple form, question answering can be incredibly challenging. Consider answering the question "Who was Adam Smith's wife?' from the Web. There exists the following snippet from a reputable website *"Smith was born in Kirkcaldy, ... (skipped 35 words) ... In 1720, he married Margaret Douglas"*. An ideal system needs to identify that "Smith" in this text is equivalent to "Adam Smith" in the question; "he" is referencing "Smith"; and text expressions of the form "X married Y" answer questions of the form "Who was X's wife?". There are three main challenges in this process:

**Scalability** A typical QA system, such as Watson (Ferrucci et al., 2010) or any of the commercial search engines, processes millions or even billions of documents for answering a question. Yet the response time is restricted to a few seconds or even fraction of seconds. Answering any possible question that is answerable from a large corpus with limited time means that the information need to be organized and indexed for fast access and reasoning.

**Representation** A fundamental building block for text understanding is paraphrasing, e.g., knowing that "X married Y" leads to "X's wife is Y". By observing users' interactions with a search engine a system may capture certain equivalence relationships among expressions in questions (Baker, 2010). However, given these observations, there is still a wide range of choices for how the meaning of expressions can be represented. *Open information extraction* approaches (Angeli et al., 2015) represent expressions by themselves, and rely on corpus statistics to calculate their similarities. This approach leads to data sparsity, and brittleness on out-of-domain text. *Vector space* approaches (Mikolov et al., 2013; Weston et al., 2014; Neelakantan et al., 2015; Miller et al., 2016) embeds text expressions into latent *continuous* spaces. They allow soft matching of semantics for arbitrary expressions, but are hard to scale to knowledge intensive tasks, which require inference with large amount of data.

**Inference** The most basic form of reasoning is to combine pieces of information and form a new piece, for example, from *co-reference*("He", "Adam Smith") and *has_spouse*("He", "Margaret Douglas") to *has_spouse*("Adam Smith", "Margaret Douglas"). As the number of pieces that need to be put together grows the search space grows exponentially – making it a hard search problem (Lao et al., 2011). Good representations dramatically reduce the search space.

Among these challenges, scalability is the main blocker to a practical end-to-end solution. Now we present our new framework, which meets all three challenges described above.

## 1.2 N-GRAM MACHINES: A SCALABLE END-TO-END APPROACH

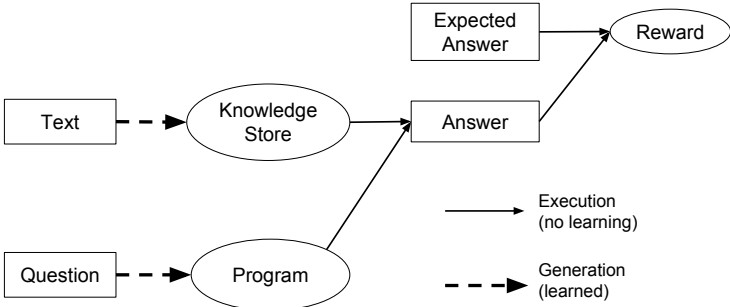

Figure 1: End-to-end QA system with a symbolic knowledge store.

We propose to solve the scalability issue of DNN text understanding models by learning to represent the meaning of text as a *symbolic* knowledge storage. Because the storage can be indexed before being used for question answering, the inference step can be done very efficiently with complexity that is

independent of the original text size. More specifically the structured storage we consider is simply a set of "n-grams", which we show can represent complex semantics presented in bAbI tasks (Weston et al., 2015) and can be indexed for efficient computations at scale. Each n-gram consists of a sequence of tokens, and each token can be a word, or any predefined special symbol. Different from conventional n-grams, which are contiguous chunks of text, the "n-grams" considered here can be any combination of arbitrary words and symbols. The whole system (Figure 1) consists of learnable components which convert text into symbolic knowledge storage and questions into programs (details in Section 2.1). A deterministic executor executes the programs against the knowledge storage and produces answers. The whole system is trained end-to-end with no human annotation other than the expected answers to a set of question-text pairs.

Training highly expressive discrete latent variable models on large datasets is a challenging problem due to the difficulties posed by inference (Hinton et al., 2006; Mnih & Gregor, 2014) – specifically the huge variance in the gradient estimation. Mnih & Gregor (2014) applies REINFORCE (Williams, 1992) to optimize a variational lower-bound of the data log-likelihood, but relies on complex schemes to reduce variance in the gradient estimation. We use a different set of techniques to learn N-Gram Machines, which are simpler and with less model assumptions. Instead of Monte Carlo integration, which is known for high variance and low data efficiency, we apply *beam search*. Beam search is very effective for deterministic environments with sparse reward (Liang et al., 2017; Guu et al., 2017), but it leads to a search problem. At inference time, since only a few top hypotheses are kept in the beam, search could get stuck and not receive any reward, preventing learning. We solve this hard search problem by 1) having an stabilized auto-encoding objective to bias the knowledge encoder to more interesting hypotheses; and 2) having a structural tweak procedure which retrospectively corrects the inconsistency among multiple hypotheses so that reward can be achieved (See Section 2.2).

## 2 FRAMEWORK

In this section we first describe the N-Gram Machine (NGM) model structure, which contains three sequence to sequence modules, and an executor that executes programs against knowledge storage. Then we describe how this model can be trained end-to-end with reinforcement learning. We use the bAbI dataset (Weston et al., 2015) as running examples.

### 2.1 NGM MODEL STRUCTURE

**Knowledge storage**  Given a sequence of sentences $\mathbf{s} = \{s_1, \ldots, s_T\}$, our knowledge storage is a collection of knowledge tuples $\mathbf{\Gamma} = \{\Gamma_1, \ldots, \Gamma_T\}$. The tuple $\Gamma_i$ has two parts: a time stamp $i$ and a sequence of symbols $(\gamma_1, \ldots, \gamma_N)$, where each symbol $\gamma_j$ is either a word from the sentence $s_i$ or a special symbol from a pre-defined set. The time stamps in the tuple are useful for reasoning about time and are just sentence indices for the bAbI task. The knowledge storage is probabilistic – each tuple $\Gamma_i$ also has a probability, and the probability of the knowledge storage is the product of the probabilities of all its tuples (Equation 1). An example of a knowledge storage is shown in Table 1.

Table 1: Example of probabilistic knowledge storage. Each sentence may be converted to a distribution over multiple tuples, but only the one with the highest probability is shown here.

| Sentences | Knowledge tuples | | |
|---|---|---|---|
| | Time stamp | Symbols | Probability |
| Mary went to the kitchen. | 1 | `mary to kitchen` | 0.9 |
| Mary picked up the milk. | 2 | `mary the milk` | 0.4 |
| John went to the bedroom. | 3 | `john to bedroom` | 0.7 |
| Mary journeyed to the garden. | 4 | `mary to garden` | 0.8 |

**Programs**  Programs in the N-Gram Machine are similar to those introduced in Neural Symbolic Machine (Liang et al., 2017), except that our functions operate on n-grams (i.e. knowledge tuples)[1] instead of Freebase triples. In general, functions specify how symbols can be retrieved from a

---

[1]We will use "n-gram" and "knowledge tuple" interchangeably.

knowledge storage. Specifically, a function in NGM use a prefix (or suffix) to retrieve symbols from tuples – i.e. if a prefix "matches" a tuple, the immediate next symbol in the tuple is returned. For the bAbI tasks, we define four functions, which are illustrated in Table 2: Function `Hop` and `HopFR` return symbols from all the matched tuples while function `Argmax` and `ArgmaxFR` return symbols from the latest matches (i.e. the tuples with the latest the time stamp in all the matches)

Table 2: Functions in N-Gram Machines. The knowledge storage on which the programs can execute is $\mathbf{\Gamma}$, and a knowledge tuple $\Gamma_i$ is represented as $(i, (\gamma_1, \ldots, \gamma_N))$. "FR" means *from right*.

| Name | Inputs | Return |
|---|---|---|
| `Hop` | $v_1 \ldots v_L$ | $\{\gamma_{L+1} \mid \text{if } (\gamma_1 \ldots \gamma_L) == (v_1, \ldots, v_L), \forall \Gamma \in \mathbf{\Gamma}\}$ |
| `HopFR` | $v_1 \ldots v_L$ | $\{\gamma_{N-L} \mid \text{if } (\gamma_{N-L+1} \ldots \gamma_N) == (v_L, \ldots, v_1), \forall \Gamma \in \mathbf{\Gamma}\}$ |
| `Argmax` | $v_1 \ldots v_L$ | $\text{argmax}_i\{(\gamma_{L+1}, i) \mid \text{if } (\gamma_1 \ldots \gamma_L) == (v_1, \ldots, v_L), \forall \Gamma_i \in \mathbf{\Gamma}\}$ |
| `ArgmaxFR` | $v_1 \ldots v_L$ | $\text{argmax}_i\{(\gamma_{N-L}, i) \mid \text{if } (\gamma_{N-L+1} \ldots \gamma_N) == (v_L, \ldots, v_1), \forall \Gamma_i \in \mathbf{\Gamma}\}$ |

More formally a program $C$ is a list of expressions $c_1...c_N$, where each $c_i$ is either a special expression `Return` indicating the end of the program, or is of the form of $(F, A_1...A_L)$ where $F$ is a function in Table 2 and $A_1...A_L$ are $L$ input arguments of $F$. When an expression is executed, it returns a set of symbols by matching its arguments in $\mathbf{\Gamma}$, and stores the result in a new variable symbol (e.g., `V1`) to reference the result. Consider executing the expression (`Hop V1 to`) against the knowledge storage in Table 1, assuming that `V1` is a variable that stores {`mary`} from previous executions. The execution returns a set of two symbols {`kitchen`, `garden`}. Similarly, executing (`Argmax V1 to`) would instead produces {`garden`}. Though executing a program on a knowledge storage as described above is deterministic, probabilities are assigned to the execution results, which are the products of probabilities of the corresponding program and knowledge storage. Since the knowledge storage can be indexed using data structures such as hash tables, the program execution time is independent of the size of the knowledge storage.

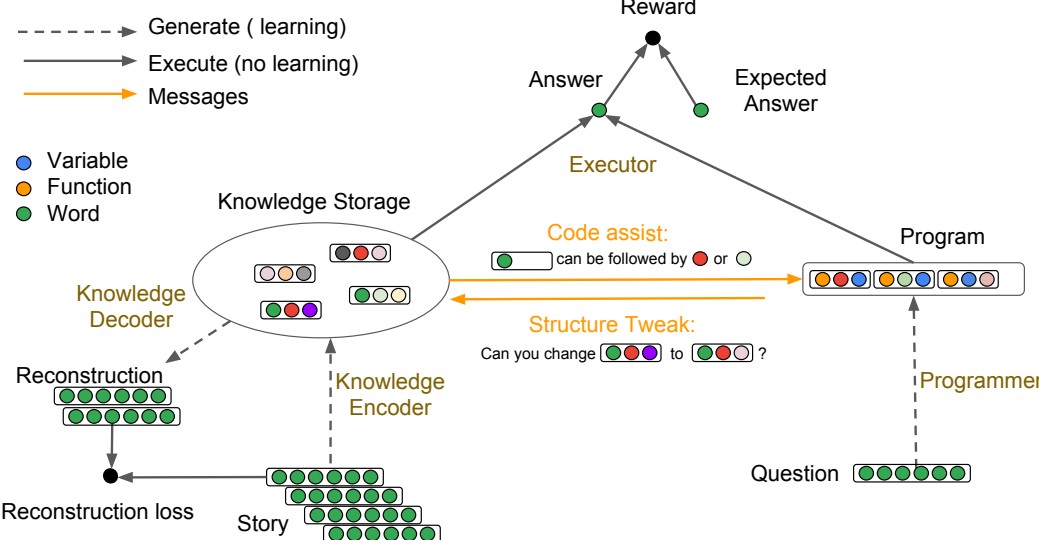

Figure 2: N-Gram Machine. The model contains two discrete hidden structures, the knowledge storage and the program, which are generated from the story and the question respectively. The executor executes programs against the knowledge storage to produce answers. The three learnable components, knowledge encoder, knowledge decoder, and programmer, are trained to maximize the answer accuracy as well as minimize the reconstruction loss of the story. Code assist and structure tweak help the knowledge encoder and programmer to communicate and cooperate with each other.

**Seq2Seq components** Our N-Gram Machine uses three sequence-to-sequence (Sutskever et al., 2014) neural network models to define probability distributions over knowledge tuples and programs. As illustrated in Figure 2, these models are:

- A *knowledge encoder* that converts sentences to knowledge tuples and defines a distribution $P(\Gamma_i|s_i, s_{i-1}; \theta_{\text{enc}})$. It is conditioned on the previous sentence $s_{i-1}$ to handle cross sentence linguistic phenomenons such as co-references[2]. The probability of a knowledge storage $\mathbf{\Gamma} = \{\Gamma_1 \ldots \Gamma_n\}$ defined as the product of its knowledge tuples' probabilities:

$$P(\mathbf{\Gamma}|\mathbf{s}; \theta_{\text{enc}}) = \Pi_{\Gamma_i \in \mathbf{\Gamma}} P(\Gamma_i|s_i, s_{i-1}; \theta_{\text{enc}}) \tag{1}$$

- A *knowledge decoder* that converts tuples back to sentences and defines a distribution $P(s_i|\Gamma_i, s_{i-1}; \theta_{\text{dec}})$. It enables auto-encoding training, which is crucial for finding good knowledge representations (See Section 2.3).

- A *programmer* that converts questions to programs and defines a distribution $P(C|q, \mathbf{\Gamma}; \theta_{\text{prog}})$. It is conditioned on the knowledge storage $\mathbf{\Gamma}$ for code assistance (Liang et al., 2017) – before generating each token the programmer can query $\mathbf{\Gamma}$ about the valid next tokens given a tuple prefix, and therefore avoid writing invalid programs.

For all of these neural networks we use the CopyNet (Gu et al., 2016) architecture, which has copy (Vinyals et al., 2015) and attention (Bahdanau et al., 2014) mechanisms. The programmer is also enhanced with a key-variable memory (Liang et al., 2017) for compositing semantics.

## 2.2 BEAM SEARCH

Given an example $(\mathbf{s}, q, a)$ from our training set, we would like to maximize the expected reward

$$O^{QA}(\theta_{\text{enc}}, \theta_{\text{prog}}) = \sum_{\mathbf{\Gamma}} \sum_{C} P(\mathbf{\Gamma}|\mathbf{s}; \theta_{\text{enc}}) P(C|q, \mathbf{\Gamma}; \theta_{\text{prog}}) R(\mathbf{\Gamma}, C, a), \tag{2}$$

where the reward function $R(\cdot)$ returns 1 if and only if executing $C$ on $\mathbf{\Gamma}$ produces $a$. We approximate the expectation with beam searches. More specifically, the summation over all programs is replaced by summing over programs found by beam search based on the programmer model $P(C|q, \mathbf{\Gamma}; \theta_{\text{prog}})$. For the summation over knowledge storages $\mathbf{\Gamma}$, we first run beam search for each sentence based on the knowledge encoder model $P(\Gamma_i|s_i, s_{i-1}; \theta_{\text{enc}})$, and then sample a set of knowledge storages by independently sampling from the knowledge tuples of each sentence. However, since the beam searches explore exponentially large spaces, it is very challenging to optimize $O^{QA}$. We introduce two special techniques to iteratively reduce and improve the search space:

**Stabilized Auto-Encoding (AE)** We add an auto-encoding objective to our framework, similar to the text summarization model proposed by Miao & Blunsom (2016). The training of this objective can be done by variational inference (Kingma & Welling, 2014; Mnih & Gregor, 2014):

$$O^{\text{VAE}}(\theta_{\text{enc}}, \theta_{\text{dec}}) = \mathbb{E}_{p(z|x; \theta_{\text{enc}})}[\log p(x|z; \theta_{\text{dec}}) + \log p(z) - \log p(z|x; \theta_{\text{enc}})], \tag{3}$$

where $x$ is text, and $z$ is the hidden discrete structure. However, it suffers from instability due to the strong coupling between encoder and decoder – the training of the decoder $\theta_{\text{dec}}$ relies solely on a distribution parameterized by the encoder $\theta_{\text{enc}}$, which changes throughout the course of training.

To improve the auto-encoding training stability, we propose to augment the decoder training with a more stable objective – predict the data $x$ back from noisy partial observations of $x$, which are independent of $\theta_{\text{enc}}$. More specifically, for NGM we force the knowledge decoder to decode from a fixed set of hidden sequences $z \in \mathbf{Z}^N(x)$, which includes all knowledge tuples of length $N$ and consist of only words from the text $x$:

$$O^{\text{AE}}(\theta_{\text{enc}}, \theta_{\text{dec}}) = \mathbb{E}_{p(z|x; \theta_{\text{enc}})}[\log p(x|z; \theta_{\text{dec}})] + \sum_{z \in \mathbf{Z}^N(x)} \log p(x|z; \theta_{\text{dec}}), \tag{4}$$

The knowledge decoder $\theta_{\text{dec}}$ converts knowledge tuples back to sentences and the reconstruction log-likelihoods approximate how informative the tuples are, which can be used as reward for the knowledge encoder. We also drop the KL divergence (last two terms in Equation 3) between language model $p(z)$ and the encoder, since the $z$'s are produced for NGM computations instead of human reading, and do not need to be fluent in natural language.

---

[2] Ideally it should condition on the partially constructed $\mathbf{\Gamma}$ at time $t - 1$, but that makes it hard to do batch training of the DNN models, and is beyond the scope of this work.

**Structure Tweaking (ST)** Even with AE training, the knowledge encoder is encoding tuples without the understanding of how they are going to be used, and may encode them inconsistently across sentences. At the later QA stage, such inconsistency can lead to no reward when the programmer tries to reason with multiple knowledge tuples. To retrospectively correct the inconsistency in tuples, we apply *structure tweak*, a procedure which is similar to code assist (Chen et al., 2017), but works in an opposite direction. While code assist uses the knowledge storage to inform the programmer, structure tweak adjusts the knowledge encoder to cooperate with an uninformed programmer.

More specifically, at training the programmer always performs an extra beam search with code assist turned off. If such programs lead to execution failure, they can be used to propose tweaked knowledge tuples (Algorithm 1). For example, when executing an expression (`Hop mary journeyed`) – generated without code assist – on a knowledge storage with tuples (`john the milk`), (`mary went bathroom`) and (`mary picked milk`), matching the prefix (`mary journeyed`) fails at token `journeyed` and returns empty result. At this point, the programmer uses `journeyed` to replace inconsistent symbols in partially matched tuples (i.e. tuples that start with `mary`), and produces tuples (`mary journeyed bathroom`) and (`mary journeyed milk`). These tweaked tuples are then added into the beams for the knowledge encoder. In this way, the search space of the knowledge storage is refined, and the knowledge encoder can learn to generate tuples in a consistent fashion in the future.

---

**Algorithm 1** Structure tweak procedure. Here we assume that the function $f$ is one of `Hop` or `Argmax`. The procedure for `HopFR` and `ArgmaxFR` can be defined similarly.

---

**Input:** Knowledge storage $\Gamma$. Expression $(f a_1 \ldots a_L)$ from an uninformed programmer.
**Output:** Tweaked knowledge tuples $\mathcal{T}$.
**if** $(a_1 \ldots a_L)$ can be matched in $\Gamma$, or $(a_1)$ can not be matched in $\Gamma$ **then**
  **return**
Let $p = (a_1, \ldots, a_m)$ be the longest prefix matched in $\Gamma$, and $\Gamma_p$ be the set of matched tuples.
**for** $\Gamma = (\gamma_1, \ldots, \gamma_N) \in \Gamma_p$ **do**
  Add $(a_1, \ldots, a_m, a_{m+1}, \gamma_{m+2} \ldots, \gamma_N)$ to $\mathcal{T}$

---

## 2.3 PARAMETERS ESTIMATION

Now the whole model has parameters $\theta = [\theta_{\text{enc}}, \theta_{\text{dec}}, \theta_{\text{prog}}]$, and the training objective function is

$$O(\theta) = O^{AE}(\theta_{\text{enc}}, \theta_{\text{dec}}) + O^{QA}(\theta_{\text{enc}}, \theta_{\text{prog}}) \tag{5}$$

$$= \sum_{s_i \in \mathbf{s}} \sum_{\Gamma_i} P(\Gamma_i | s_i, s_{i-1}; \theta_{\text{enc}})[\beta(\Gamma_i) + \log P(s_i | \Gamma_i, s_{i-1}; \theta_{\text{dec}})] \tag{6}$$

$$+ \sum_{\mathbf{\Gamma}} \sum_{C} P(\mathbf{\Gamma} | \mathbf{s}; \theta_{\text{enc}}) P(C | q, \mathbf{\Gamma}; \theta_{\text{prog}}) R(\mathbf{\Gamma}, C, a) \tag{7}$$

where $\beta(\Gamma_i)$ is 1 if $\Gamma_i$ only contains tokens from $s_i$ and 0 otherwise.

For training stability and to overcome search failures, we augment this objective with experience replay (Schaul et al., 2016), and the gradients with respect to each set of parameters are:

$$\nabla_{\theta_{\text{dec}}} O'(\theta) = \sum_{s_i \in \mathbf{s}} \sum_{\Gamma_i} [\beta(\Gamma_i) + P(\Gamma_i | s_i, s_{i-1}; \theta_{\text{enc}})] \nabla_{\theta_{\text{dec}}} \log P(s_i | \Gamma, s_{i-1}; \theta_{\text{dec}}), \tag{8}$$

$$\nabla_{\theta_{\text{enc}}} O'(\theta) = \sum_{s_i \in \mathbf{s}} \sum_{\Gamma_i} [P(\Gamma_i | s_i, s_{i-1}; \theta_{\text{enc}}) \log P(s_i | \Gamma_i, s_{i-1}; \theta_{\text{dec}}) \tag{9}$$

$$+ \mathcal{R}(\mathcal{G}'(\Gamma_i)) + \mathcal{R}(\mathcal{G}(\Gamma_i))] \nabla_{\theta_{\text{enc}}} \log P(\Gamma_i | s_i, s_{i-1}; \theta_{\text{enc}}), \tag{10}$$

where $\mathcal{R}(\mathcal{G}) = \sum_{\mathbf{\Gamma} \in \mathcal{G}} \sum_{C} P(\mathbf{\Gamma} | \mathbf{s}; \theta_{\text{enc}}) P(C | q, \mathbf{\Gamma}; \theta_{\text{prog}}) R(\mathbf{\Gamma}, C, a)$ is the total expected reward for a set of valid knowledge stores $\mathcal{G}$, $\mathcal{G}(\Gamma_i)$ is the set of knowledge stores which contains the tuple $\Gamma_i$, and $\mathcal{G}'(\Gamma_i)$ is the set of knowledge stores which contains the tuple $\Gamma_i$ through tweaking.

$$\nabla_{\theta_{\text{prog}}} O'(\theta) = \sum_{\mathbf{\Gamma}} \sum_{C} [\alpha I [C \in \mathcal{C}^*(\mathbf{s}, q)] + P(C | q, \mathbf{\Gamma}; \theta_{\text{prog}})] \tag{11}$$

$$\cdot P(\mathbf{\Gamma} | \mathbf{s}; \theta_{\text{enc}}) R(\mathbf{\Gamma}, C, a) \nabla_{\theta_{\text{prog}}} \log P(C | q, \mathbf{\Gamma}; \theta_{\text{prog}}), \tag{12}$$

where $\mathcal{C}^*(\mathbf{s}, q)$ is the experience replay buffer for $(\mathbf{s}, q)$. $\alpha = 0.1$ is a constant. During training, the program with the highest weighted reward (i.e. $P(\boldsymbol{\Gamma}|\mathbf{s}; \theta_{\text{enc}})R(\boldsymbol{\Gamma}, C, a)$) is added to the replay buffer.

Because the knowledge storage and the program are non-differentiable discrete structures, we optimize our objective by a coordinate ascent approach – optimizing the three components in alternation with REINFORCE (Williams, 1992).

## 3 RESULTS

We apply the N-Gram Machine (NGM) to solve a set of text reasoning tasks in the bAbI dataset (Weston et al., 2015). In Section 3.1, we demonstrate that the model can learn to build knowledge storage and generate programs that accurately answer the questions. In Section 3.2, we show the scalability advantage of NGMs by applying it to longer stories up to 10 million sentences.

The Seq2Seq components are implemented as one-layer recurrent neural networks with Gated Recurrent Unit (Chung et al., 2014). The hidden dimension and the vocabulary embedding dimension are both 8. During decoding, the per sentence knowledge tuple beam size is 2, the knowledge store sample size is 5, and the program beam size is 30. The neural networks are implemented in TensorFlow (Abadi et al., 2016) and optimized using Adam (Kingma & Ba, 2014).

### 3.1 BABI

The bAbI dataset contains twenty tasks in total. We consider the subset of them that are extractive question answering tasks (as defined in Section1.1). Each task is learned separately. For all tasks, we set the knowledge tuple length to three. In Table 3, we report results on the test sets. NGM outperforms MemN2N (Sukhbaatar et al., 2015) on all tasks listed. The results show that auto-encoding is essential to bootstrapping the learning. Without auto-encoding, the expected rewards are near zero; but auto-encoding alone is not sufficient to achieve high rewards (See Section 2.2). Since multiple discrete latent structures (i.e. knowledge tuples and programs) need to agree with each other over the choice of their representations for QA to succeed, the search becomes combinatorially hard. Structure tweaking is an effective way to refine the search space – improving the performance of more than half of the tasks.

Table 3: Test accuracy on bAbI tasks with auto-encoding (AE) and structure tweak (ST)

|  | Task 1 | Task 2 | Task 11 | Task 15 | Task 16 |
|---|---|---|---|---|---|
| MemN2N | 1.000 | 0.830 | 0.840 | 1.000 | 0.440 |
| QA | 0.007 | 0.027 | 0.000 | 0.000 | 0.098 |
| QA + AE | 0.709 | 0.551 | **1.000** | 0.246 | **1.000** |
| QA + AE + ST | **1.000** | **0.853** | **1.000** | **1.000** | **1.000** |

To illustrate the effect of auto-encoding, we show in Figure 3 how informative the knowledge tuples are by computing the reconstruction log-likelihood using the knowledge decoder for the sentence "john went back to the garden". As expected, the tuple (`john went garden`) is the most informative. Other informative tuples include (`john the garden`) and (`john to garden`). Therefore, with auto-encoding training, useful hypotheses have large chance to be found by a small knowledge encoder beam size (2 in our case).

Figure 3: Visualization of the knowledge decoder's assessment of how informative the knowledge tuples are. Yellow means high and red means low.

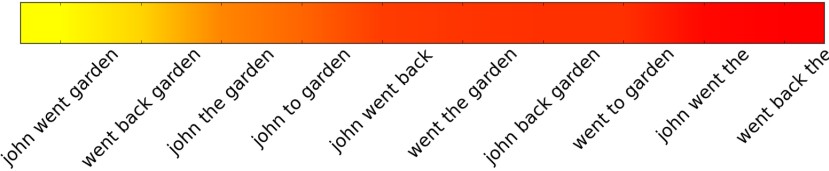

Table 4 lists sampled knowledge storages learned with different objectives and procedures. Knowledge storages learned with auto-encoding are much more informative compared to the ones without. After structure tweaking, the knowledge tuples converge to use more consistent symbols – e.g., using `went` instead of `back` or `travelled`. Our experiment results show the tweaking procedure can help NGM to deal with various linguistic phenomenons such as singular/plural ("cats" vs "cat") and synonyms ("grabbed" vs "got"). More examples are included in the supplementary material A.1.

Table 4: Sampled knowledge storage with question answering (QA) objective, auto-encoding (AE) objective, and structure tweak (ST) procedure. Using AE alone produces similar tuples to QA+AE. The differences between the second and the third column are underlined.

| QA | QA + AE | QA + AE + ST |
|---|---|---|
| `went went went` | `daniel went office` | `daniel went office` |
| `mary mary mary` | `mary` `back` `garden` | `mary` `went` `garden` |
| `john john john` | `john` `back` `kitchen` | `john` `went` `kitchen` |
| `mary mary mary` | `mary` `grabbed` `football` | `mary` `got` `football` |
| `there there there` | `sandra got apple` | `sandra got apple` |
| `cats cats cats` | `cats` `afraid wolves` | `cat` `afraid wolves` |
| `mice mice mice` | `mice` `afraid wolves` | `mouse` `afraid wolves` |
| `is is cat` | `gertrude is cat` | `gertrude is cat` |

## 3.2 LIFE-LONG BABI

To demonstrate the scalability advantage of the N-Gram Machine, we conduct experiments on question answering data where the number of sentences may increase up to 10 million. More specifically we generated longer bAbI stories using the open-source script from Facebook[3]. We measure the answering time and answer quality of MemN2N (Sukhbaatar et al., 2015)[4] and NGM at different scales. The answering time is measured by the amount of time used to produce an answer when a question is given. For MemN2N, this is the neural network inference time. For NGM, because the knowledge storage can be built and indexed in advance, this is the programmer decoding time.

Figure 4: Scalability comparison of MemN2N and NGM. Left: Answering time. Right: Answer quality. Story length is the number of sentences in each QA pair.

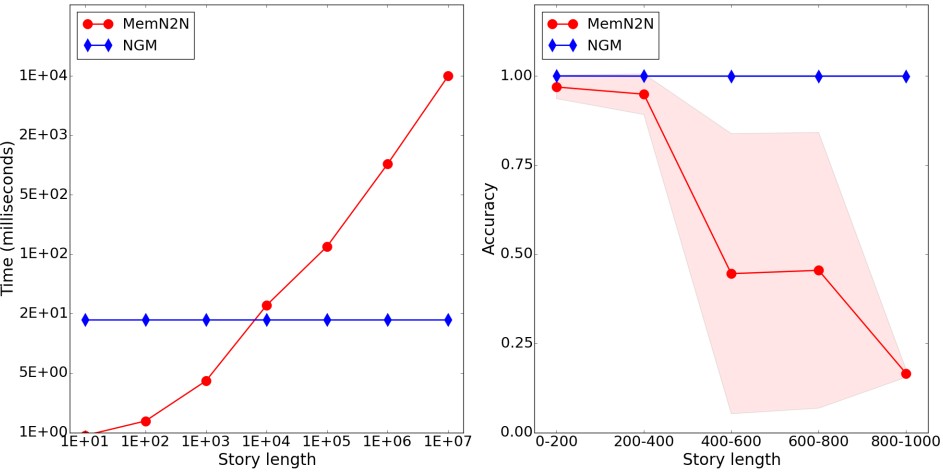

Figure 4 compares MemN2N and NGM. In terms of answering time, MemN2N scales poorly – the inference time increases linearly as the story length increases. While for NGM, the answering time is not affected by story length. The crossover of the two lines is when the story length is around 1000, which is due to the difference in neural network architectures – NGM uses recurrent networks

---

[3]`https://github.com/facebook/bAbI-tasks`
[4]`https://github.com/domluna/memn2n`

while MemN2N uses feed-forward networks. To compare the answer quality at scale, we apply MemN2N and NGM to solve three life-long bAbI tasks (Task 1, 2, and 11). For each life-long task, MemN2N is run for 10 trials and the test accuracy of the trial with the best validation accuracy is used. For NGM, we use the same models trained on regular bAbI tasks. We compute the average and standard deviation of test accuracy from these three tasks. MemN2N performance is competitive with NGM when story length is no greater than 400, but decreases drastically when story length further increases. On the other hand, NGM answering quality is the same for all story lengths. These scalability advantages of NGM are due to its "machine" nature – the symbolic knowledge storage can be computed and indexed in advance, and the program execution is robust on stories of various lengths.

## 4 RELATED WORK

The auto-encoding part of our model (Figure 2) is similar to the auto-encoding text summarization model proposed by Miao & Blunsom (2016). Miao & Blunsom (2016) solved the large search space problem (when generating hidden sequences) by 1) restricting hidden sequences to only consist of tokens in the source sequences (through a PointerNet (Vinyals et al., 2015)), and 2) a language model pre-trained on a separate corpus. We adopt a different approach, which does not require a separate corpus, or a restricted hidden sequence space: 1) use a less restricted hidden space (through a CopyNet (Gu et al., 2016)) by allowing both copied tokens and generated tokens; 2) stabilize the decoder by forcing it (through experience replay) to train from randomly generated hidden sequences; and 3) use the log-likelihood of the pre-trained decoder to guide the training of the encoder.

The question answering part of our model (Figure 2) is similar to the Neural Symbolic Machine (NSM) (Liang et al., 2017), which is a memory enhanced sequence-to-sequence model that translates questions into programs in $\lambda$-calculus (Liang et al., 2011). The programs, when executed on a knowledge graph, can produce answers to the questions. Our work extends NSM by removing the assumption of a given knowledge bases or schema, and instead learns to generate storage by end-to-end training to answer questions.

The reminder of this section analyzes why existing text understanding technologies are fundamentally limited and draws connections of our proposed solution to psychology and neuroscience.

### 4.1 TEXT UNDERSTANDING: CURRENT PRACTICE

In recent years, several large-scale knowledge bases (KBs) have been constructed, such as YAGO (Suchanek et al., 2007), Freebase (Bollacker et al., 2008), NELL (Mitchell et al., 2015), Google Knowledge Graph (Singhal, 2012), Microsoft Satori (Qian, 2013), and others. However, all of them suffer from a completeness problem (Dong et al., 2014) relative to large text corpora, which which stems from an inability to convert *arbitrary* text to graph structures, and accurately answering questions from these structures. There are two core issues in this difficulty:

**The schema (or representation) problem** Traditional text extraction approaches need some fixed and finite target schema (Riedel et al., 2013). For example, the definition of marriage[5] is very complex and includes a whole spectrum of concepts, such as common-law marriage, civil union, putative marriage, handfasting, nikah mut'ah, and so on. It is prohibitively expensive to have experts to clearly define all these differences in advance, and annotate enough utterances from which they are expressed. However, given a particular context, many of this distinctions might be irrelevant. For example when a person say "I just married Lisa." and then ask "Who is my wife?", a system should be able to respond correctly, without first learning about all types of marriage. A desirable solution should induce schema automatically from the corpus such that knowledge encoded with this schema can help downstream tasks such as QA.

**The end-to-end training (or inference) problem** The state-of-the-art approaches break down QA solutions into independent components, such as schema definition and manually annotating examples (Mitchell et al., 2015; Bollacker et al., 2008), relation extraction (Mintz et al., 2009), entity resolution (Singla & Domingos, 2006), and semantic parsing of questions (Berant et al., 2013; Liang

---

[5]https://en.wikipedia.org/wiki/Marriage

et al., 2017). However, these components need to work together and depend on each other. For example, the meaning of text expressions (e.g., "father of IBM", "father of Adam Smith", "father of electricity") depend on entity types. The type information, however, depends on the entity resolution decisions, and the content of the knowledge base (KB). The content of the KBs, if populated from text, will in return depend on the meaning of text expressions. Existing systems (Dong et al., 2014; Berant et al., 2013) leverage human annotation and supervised training for each component separately, which create consistency issues, and preclude directly optimizing the performance of the QA task. A desirable solution should use few human annotations of intermediate steps, and rely on end-to-end training to directly optimize the QA quality.

## 4.2 Text Understanding: Deep Neural Nets

More recently there has been a lot of progress in applying deep neural networks (DNNs) to text understanding (Mikolov et al., 2013; Weston et al., 2014; Neelakantan et al., 2015; Graves et al., 2016; Miller et al., 2016). The key ingredient to these solutions is embedding text expressions into a latent *continuous* space. This removes the need to manually define a schema, while the vectors still allow complex inference and reasoning. Continuous representations greatly simplified the system design of QA systems, and enabled end-to-end training, which directly optimizes the QA quality.

However, a key issue that prevents these models from being applied to many applications is scalability. After receiving a question, all text in a corpus need to be analyzed by the model. Therefore it leads to at least $O(n)$ complexity, where $n$ is the text size. Approaches which rely on a search subroutine (e.g., DrQA (Chen et al., 2017)) lose the benefit of end-to-end training, and are limited by the quality of the retrieval stage, which itself is difficult.

## 4.3 Relationship to Psychology and Neuroscience

Previous studies (Wickman, 2012; Bartol et al., 2015) of hippocampus suggests that human memory capacity may be somewhere between 10 terabytes and 100 terabytes. This huge capacity is very advantageous for humans' survival, yet puts human brain in the same situation of a commercial search engine – facing the task of organizing vast amounts of information in a form which support fast retrieval and inference.

Our choice of using n-grams as the unit of knowledge storage is partially motivated by the Adaptive Resonance Theory (ART) (Carpenter & Grossberg, 2003; Tan et al., 2007; Wang et al., 2012), which models each piece of human episodic memory as a set of symbols, each representing one aspect of the experience such as location, color, object etc.

The structure of our model (Figure 2) resembles the complementary learning theory (McClelland et al., 1995; O'Reilly et al., 2014; Kumaran et al., 2016), which hypothesizes that intelligent agents possess two learning systems, instantiated in mammals in the neocortex and hippocampus. The first gradually acquires structured knowledge representations, while the second quickly learns the specifics of individual experiences. In our model the stories in training examples (episodic memories) and knowledge store (semantic memory) represents the fast learning neurons (declarative memory), while the sequence to sequence DNN models represent the slow learning neurons (implicit memory). The DNN training procedure, which goes over the past experience (stories) over and over again, resembles the "replay" of hippocampal memories that allows goal-dependent weighting of experience statistics (Kumaran et al., 2016).

The structure tweak procedure (Section 2.2) in our model is critical for its success, and bears resemblance to the reconstructive (or generative) memory theory (Piaget, 1977; Block, 1982), which hypothesizes that by employing reconstructive processes, individuals supplement other aspects of available personal knowledge into the gaps found in episodic memory in order to provide a fuller and more coherent version. Our analysis of the QA task showed that at knowledge encoding stage information is often encoded without understanding how it is going to be used. So it might be encoded in an inconsistent way. At the later QA stage an inference procedure tries to derive the expected answer by putting together several pieces of information, and fails due to inconsistency. Only at that time can a hypothesis be formed to retrospectively correct the inconsistency in memory. These "tweaks" in memory later participate in training the knowledge encoder in the form of experience replay.

## 5    CONCLUSIONS AND FUTURE WORK

We present an end-to-end trainable system which combines an text auto-encoding component for encoding the meaning of text in symbolic representations, and a memory enhanced sequence-to-sequence component for answering questions from the storage. We show that the method achieves good scaling properties and robust inference on artificially generated stories of up to 10 million sentences long. The system we present here illustrates how end-to-end learning and scalability can be made possible through a symbolic knowledge storage.

To further improve the system, we are interested in investigating whether the proposed n-gram representation is sufficient for natural languages. More complex representations, such as Abstract Meaning Representations Banarescu et al. (2013), are possible alternatives, but it remains unclear how to design effective weakly supervised learning techniques to induce such representations.

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

## A  SUPPLEMENTARY MATERIAL

### A.1  MODEL GENERATED KNOWLEDGE STORAGES AND PROGRAMS FOR BABI TASKS

The following tables show one example solution for each type of task. Only the tuple with the highest probability is shown for each sentence.

Table 5: Task 1 Single Supporting Fact

| Story | Knowledge Storage |
|---|---|
| Daniel travelled to the office. | Daniel went office |
| John moved to the bedroom. | John went bedroom |
| Sandra journeyed to the hallway. | Sandra went hallway |
| Mary travelled to the garden. | Mary went garden |
| John went back to the kitchen. | John went kitchen |
| Daniel went back to the hallway. | Daniel went hallway |
| **Question** | **Program** |
| Where is Daniel? | Argmax Daniel went |

Table 6: Task 2 Two Supporting Facts

| Story | Knowledge Storage |
|---|---|
| Sandra journeyed to the hallway. | Sandra journeyed hallway |
| John journeyed to the bathroom. | John journeyed bathroom |
| Sandra grabbed the football. | Sandra got football |
| Daniel travelled to the bedroom. | Daniel journeyed bedroom |
| John got the milk. | John got milk |
| John dropped the milk. | John got milk |
| **Question** | **Program** |
| Where is the milk? | ArgmaxFR milk got |
|  | Argmax V1 journeyed |

Table 7: Task 11 Basic Coreference

| Story | Knowledge Storage |
|---|---|
| John went to the bathroom. | John went bathroom |
| After that he went back to the hallway. | John he hallway |
| Sandra journeyed to the bedroom | Sandra Sandra bedroom |
| After that she moved to the garden | Sandra she garden |
| **Question** | **Program** |
| Where is Sandra? | Argmax Sandra she |

Table 8: Task 15 Basic Deduction

| Story | Knowledge Storage |
|---|---|
| Sheep are afraid of cats. | Sheep afraid cats |
| Cats are afraid of wolves. | Cat afraid wolves |
| Jessica is a sheep. | Jessica is sheep |
| Mice are afraid of sheep. | Mouse afraid sheep |
| Wolves are afraid of mice. | Wolf afraid mice |
| Emily is a sheep. | Emily is sheep |
| Winona is a wolf. | Winona is wolf |
| Gertrude is a mouse. | Gertrude is mouse |
| **Question** | **Program** |
| What is Emily afraid of? | Hop Emily is |
|  | Hop V1 afraid |

Table 9: Task 16 Basic Induction

| Story | Knowledge Storage |
|---|---|
| Berhard is a rhino. | Bernhard a rhino |
| Lily is a swan. | Lily a swan |
| Julius is a swan. | Julius a swan |
| Lily is white. | Lily is white |
| Greg is a rhino. | Greg a rhino |
| Julius is white. | Julius is white |
| Brian is a lion. | Brian a lion |
| Bernhard is gray. | Bernhard is gray |
| Brian is yellow. | Brian is yellow |
| **Question** | **Program** |
| What color is Greg? | Hop Greg a |
|  | HopFR V1 a |
|  | Hop V2 is |

