# OpenReview forum: "LEARNING TO ORGANIZE KNOWLEDGE WITH N-GRAM MACHINES"
_ICLR.cc/2018/Conference — Invite to Workshop Track_

### Official Review · AnonReviewer3 · 2017-11-27
**Lacks proper evaluation and sufficient novelty**

**Rating:** 4
**Confidence:** 4

**Review:**

The authors propose the N-Gram machine to answer questions over long documents. The model first encodes the document via tuple extraction. An autoencoder objective is used to produce meaningful tuples. Then, the model generates a program, based on the extracted tuple collection and the question, to find an answer.

I am very disappointed in the authors' choice of evaluation, namely bAbI - a toy, synthetic task long abandoned by the NLP community because of its lack of practicality. If the authors would like to demonstrate question answering on long documents, they have the luxury of choosing amongst several large scale, realistic question answering datasets such as the Stanford Question answering dataset or TriviaQA.
Beyond the problem of evaluation, the model the authors propose does not provide new ideas, and rather merges existing ones. This, in itself, is not a problem. However, the authors decline to cite many, many important prior work. For example, the tuple extraction described by the authors has significant prior work in the information retrieval community (e.g. knowledge base population, relation extraction). The idea of generating programs to query over populated knowledge bases, again, has significant related work in semantic parsing and program synthesis. Question answering over (much more complex) probabilistic knowledge graphs have been proposed before as well (in fact I believe Matt Gardner wrote his entire thesis on this topic). Finally, textual question answering (on realistic datasets) has seen significant breakthroughs in the last few years. Non of these areas, with the exception of semantic parsing, are addressed by the author. With sufficient knowledge of related works from these areas, I find that the authors' proposed method lacks proper evaluation and sufficient novelty.

---

> ### Author Response · Authors · 2017-12-18
> **Response**
>
> We thank the reviewer for the insightful feedback.
>
> [lack of sufficient novelty and missing citations]
> We disagree with the reviewer and would like to clarify the novelty of our proposed framework. The novelty in our framework is the end-to-end objective function (Equation 2), which learns to construct knowledge storage using down-stream QA tasks as weak supervision. This objective function is different from the ones in the related work mentioned by the reviewer. More specifically, 1) comparing to relation extraction, our method does not use expert-defined schema as supervision; 2) comparing to QA over knowledge graph, our method does not assume knowledge graph is given and instead constructs knowledge storage from text.
> About the reading comprehension tasks (e.g., SQUAD), they are not comparable to our work since they do not need to solve the search (from a big corpus) problem.
>
> [lack of evaluation]
> We understand the disappointment about evaluation. At this point, we can only defend that this is a theoretic work, which proposes a novel framework, and points to a new direction of how a long lasting problem in search might be solved.

---

### Official Review · AnonReviewer1 · 2017-11-29

**Rating:** 5
**Confidence:** 4

**Review:**

This paper presents the n-gram machine, a model that encodes sentences into simple symbolic representations ("n-grams") which can be queried efficiently. The authors propose a variety of tricks (stabilized autoencoding, structured tweaking) to deal with the huge search space, and they evaluate NGMs on five of the 20 bAbI tasks. I am overall a fan of the general idea of this paper; scaling up to huge inputs is definitely a necessary research direction for QA. However, I have some concerns about the specific implementation and model discussed here. How much of the proposed approach is specific to getting good results on bAbI (e.g., conditioning the knowledge encoder on only the previous sentence, time stamps in the knowledge tuple, super small RNNs, four simple functions in the n-gram machine, structure tweaking) versus having a general-purpose QA model for natural language? Addressing some of these issues would likely prevent scaling to millions of (real) sentences, as the scalability is reliant on programs being efficiently executed (by simple string matching) against a knowledge storage. The paper is missing a clear analysis of NGM's limitations... the examples of knowledge storage from bAbI in the supplementary material are also underwhelming as the model essentially just has to learn to ignore stopwords since the sentences are so simple. In its current form, I am borderline but leaning towards rejecting this paper.

Other questions:
- is "n-gram" really the most appropriate term to use for the symbolic representation? N-grams are by definition contiguous sequences... The authors may want to consider alternatives.
- why focus only on extractive QA? The evaluations are only conducted on 5 of the 20 bAbI tasks, so  it is hard to draw any conclusions from the results as to the validity of this approach. Can the authors comment on how difficult it will be to add functions to the list in Table 2 to handle the other 15 tasks? Or is NGM strictly for extractive QA?
- beam search is performed on each sentence in the input story to obtain knowledge tuples... while the answering time may not change (as shown in Figure 4) as the input story grows, the time to encode the story into knowledge tuples certainly grows, which likely necessitates the tiny RNN sizes used in the paper. How long does the encoding time take with 10 million sentences?
- Need more detail on the programmer architecture, is it identical to the one used in Liang et al., 2017?

---

> ### Author Response · Authors · 2017-12-18
> **Response**
>
> We thank the reviewer for the insightful feedback.
>
> [“N-gram” might be a misleading term]
> We agree that “N-gram” could be misleading, since it commonly means sequences of contiguous words. We are considering other names to use in the future, such as "skip n-gram", or  “engram” https://en.wikipedia.org/wiki/Engram_(neuropsychology).
>
> [why only extractive QA?]
> Extractive QA is a family of representative tasks in text understanding. To handle non-extractive QA tasks, we will need to add other functions, which operate on infinite domains (e.g., mathematical operations). The overall model structure should not change, but is beyond the scope of the current result.
>
> [How long does the encoding take with 10 million sentences?]
> With our current implementation, scoring 10M sentences would take more than two hours on a single machine without parallelization. A typical commercial search engine uses thousands of machines to encode the meaning of pages (indexing). Even with more complex LSTM structures, scalability is not likely to be an issue for encoding.
>
> [model design overfit the bAbI dataset?]
> We agree that the n-gram design and function design have limited expressiveness. We are currently working on more datasets to further understand the balance between model expressiveness and learning difficulty.

---

### Official Review · AnonReviewer2 · 2017-11-30
**An interesting, but weird framework for bAbI QA**

**Rating:** 4
**Confidence:** 3

**Review:**

The paper presents an interesting framework for bAbI QA.  Essentially, the argument is that when given a very long paragraph, the existing approaches for end-to-end learning becomes very inefficient (linear to the number of the sentences).  The proposed alternative is to encode the knowledge of each sentence symbolically as n-grams, which is thus easy to index.  While the argument makes sense, it is not clear to me why one cannot simply index the original text. The additional encode/decode mechanism seems to introduce unnecessary noise.  The framework does include several components and techniques from latest recent work, which look pretty sophisticated. However, as the dataset is generated by simulation, with a very small set of vocabulary, the value of the proposed framework in practice remains largely unproven.

Pros:
  1. An interesting framework for bAbI QA by encoding sentence to n-grams

Cons:
  1. The overall justification is somewhat unclear
  2. The approach could be over-engineered for a special, lengthy version of bAbI and it lacks evaluation using real-world data

---

> ### Author Response · Authors · 2017-12-18
> **Response**
>
> We thank the reviewer for the insightful feedback.
>
> [why not index the text directly?]
> The proposed knowledge encoder is indeed learning to index the text. From an information retrieval perspective, we expect the proposed approach to be a goal-dependent index mechanism, and produces better quality index than traditional indexing approaches. We are not aware of any existing work in this domain.
>
> [sophisticated model but simulated data]
> The model architecture is not more sophisticated than a directed probabilistic graphical model with two discrete latent variables, as shown in Figure 1. One might say that the inference procedure is complex, but this is a common challenge shared by many graphical models. The code assist and structure tweak techniques are very similar to conditional sampling (e.g. Gibbs sampling). Therefore, the proposed learning method is principled, and the choice of dataset does not affect this. We will clarify this in the final version.

---

### Decision · Program_Chairs · 2018-01-29
**ICLR 2018 Conference Acceptance Decision**

**Decision:**

Invite to Workshop Track

**Comment:**

i am a big fan of this idea, but i agree with the reviewers that evaluating this idea on bAbI (which was originally created from a small set of rules and primitives) discounts quite a bit of what is being claimed here. one of the future directions mentioned by the authors ("investigating whether the proposed n-gram representation is sufficient for natural languages") should have been included even with a negative result, which would've increased the significance significantly.